# Safety.Net: A Pilot Study on a Multi-Risk Internet Prevention Program

**DOI:** 10.3390/ijerph18084249

**Published:** 2021-04-16

**Authors:** Jéssica Ortega-Barón, Joaquín González-Cabrera, Juan M. Machimbarrena, Irene Montiel

**Affiliations:** 1Faculty of Education, Universidad Internacional de la Rioja (UNIR), Avenida de la Paz, 137, 26006 Logroño, Spain; jessica.ortega@unir.net (J.O.-B.); irene.montiel@unir.net (I.M.); 2Faculty of Psychology, Universidad del País Vasco (UPV/EHU), Avenida de Tolosa, 70, 20018 Donostia, Spain; juanmanuel.machimbarrena@ehu.eus

**Keywords:** program, prevention, evaluation, risks, internet, adolescents

## Abstract

Many programs exist to prevent bullying and cyberbullying. Nevertheless, despite evidence of the numerous overlapping risks of the Internet, programs that jointly and adequately address large sets of risks are not presently described in the scientific literature. This study’s main objective was to assess the effectiveness of the Safety.net program in a pilot sample. This program prevents eight Internet risks: cyberbullying, sexting, online grooming, cyber dating abuse, problematic Internet use, nomophobia, Internet gaming disorder, and online gambling disorder. The Safety.net program comprises 16 sessions and 4 modules (digital skills, relational risks, dysfunctional risks, and change of attitudes and cognitions). Each session lasts one hour, but the program has a networked instructional design to recall previous content in later sessions. For its assessment, a pre/post-test repeated measures design with a control group and an intervention group was used. The study sample was 165 adolescents between 11 and 14 years old (M = 12.11, SD = 0.89). The intervention group demonstrated improvements compared to the control group concerning online grooming, problematic Internet use, Internet gaming disorder, and nomophobia. These results suggest that the Safety.net program is effective in preventing the increase of most of the assessed risks and that it reduces some of them with a small number of sessions.

## 1. Introduction

Heavy and continual use of information and communication technologies (ICTs) has provided adolescents with numerous ways to communicate and interact. In fact, 80% of European adolescents ranging from 9 to 16 years of age declare that they use their smartphones daily to connect to the Internet and use social networks [1]. Although the Internet has many positive aspects for their personal and social development, it also has facilitated the emergence and continuity of many problems and online forms of violence affecting adolescents. The most obvious example of such problems in the school context is cyberbullying, which is defined as intentional violent behavior through technologies that repeatedly take place (at any time and in any place) against a victim, who cannot defend him/herself easily [2]. According to a systematic review of 159 cyberbullying studies, the prevalence of cybervictimization ranges between 1% and 61.1% [3]. Other risks are sexting, online grooming, and cyber dating abuse. Sexting refers to the act of sending sexual content (mostly photographs and/or videos), generally produced by oneself, to other people through technologies [4]. The average prevalence of this form of online behavior in adolescents and youth is 14.8% [5]. Online grooming refers to the process through, which an adult using ICTs, comes into contact with a minor and gains his/her confidence to create and/or maintain some kind of sexual interaction with him/her [6]. The average prevalence rate of online sexual solicitation is 11.5% [5]. Cyberdating abuse involves using threats, insults, humiliation, and/or denigration within the online context with the intent of isolating, controlling and causing anguish to one’s partner [7]. The prevalence of this type of victimization ranges from 5.8% to 92% [8]. In general, the wide variability in the prevalence of these Internet risks is due to the numerous methodologies, samples, and sociocultural contexts of the different studies. It should be noted that all of these risks share a common characteristic—these behaviors result from minors’ interaction with other people within the online context. Thus, the aforementioned risks (cyberbullying, cyber dating abuse, sexting, and grooming) are labeled under the term relational Internet risks.

In contrast to relational risks, other behaviors are more inherently linked to the dysfunctional use of ICTs. One of the most studied risks is problematic Internet use (PIU), which is characterized by the existence of negative consequences arising from inadequate self-regulation (compulsive use of the Internet and uneasiness when not connected), preference for online social interaction, and using the Internet to modify one’s state of mind [9]. The prevalence of problematic Internet use ranges from 14.3% to 54.9% [10]. In addition to PIU, several other specific dysfunctional uses of technology are prevalent among adolescents, such as Internet gaming disorder (IGD), online gambling disorder (OGD), and nomophobia. Internet gaming disorder refers to recurrent, persistent participation in online video games that causes clinically significant distress [11]. The prevalence of this disorder among adolescents is 4.6%, according to the meta-analysis carried out by Fam [12]. Online gambling disorder refers to persistent gambling with money that can cause clinically significant deterioration or discomfort. Among Spanish adolescents, this disorder’s prevalence is approximately 1%, and nearly 3% are at risk of suffering from it [13]. Nomophobia refers to an intense, irrational, disproportionate fear of the possibility of not being able to use one’s smartphone, and it is a prevalent problem among adolescents. In Spain, its prevalence is approximately 15% [14], although this rate varies greatly depending on the context and the assessment method employed for evaluation [15]. All of these problems are labeled as dysfunctional Internet risks because they share the excessive use of ICTs (without necessarily involving other people).

There is a shortage of literature addressing the concurrence of the aforementioned relational and dysfunctional risks. However, some studies provide evidence of overlapping relational risks among adolescents [16,17,18]. Machimbarrena et al. [19] indicated that 5.49% of adolescents showed cybervictimization due to cyberbullying, sexting, cyber dating abuse, and online grooming, and over 50% were involved in more than one risk at the same time. Furthermore, few studies have assessed all of the aforementioned dysfunctional risks; however, several have suggested that there is an important relationship between them [13,20]. Additionally, several longitudinal studies have provided evidence of a natural tendency for cybervictimization, sexting, online grooming, and cyber dating abuse to increase after one year [21,22,23].

Considering this plurality and concurrence of Internet risks, their prevention is essential, and it is necessary to promote the positive use of technologies among children and adolescents. Most previous programs intended to prevent specific Internet risks, particularly cyberbullying [24]. In Spain, two such programs are “Convivir en un mundo real y digital” [25] and “cibermentores” [26]. At the international level, the cyber-friendly schools and Kiva programs focused on cyberbullying [27,28] are of particular note. The ThinkUKnow Internet safety program [29] is focused on online grooming, and DARSI is focused on cyber dating abuse [30]. A few programs deal jointly with several relational Internet risks, such as cyberbullying, sexting, and online grooming—these include Cyberprogram 2.0 [31], the Prev@cib program [32], and Preventive Intervention on Cyberbullying and Grooming [33].

There are also very few programs with indicators of empirical effectiveness in the field of dysfunctional Internet risks. In Spain, the most prominent programs are ADITEC [34], which is focused on abuse of new technologies, and “¿Qué Te Juegas?”, which addresses gambling disorders [35]. Of note at the international level are PROTECT, for problematic Internet use [36], and the gambling prevention program [37]. Currently, there are no programs to prevent Internet gaming disorders in Spain. However, some international programs exist, such as The Game Over Intervention [38]. Regarding nomophobia, general problematic Internet use, and, particularly, online gambling disorder, there is no evidence of any scientifically validated program at the Spanish level. Despite all of the aforementioned preventive actions, no program jointly addresses all of the aforementioned risks in an adequate manner while focusing on an age range that is widely accepted as crucial for primary prevention (11–13 years old) [39].

Therefore, the present study’s main objective was to assess the effectiveness of the Safety.net program in a pilot sample of adolescents. Due to the multi-risk approach of the program, with one risk addressed per session, the incidence of relational risks (cyberbullying, sexting, online grooming, and cyber dating abuse) and dysfunctional risks (problematic Internet use, Internet gaming disorder, and nomophobia) were not expected to increase among the intervention group. In the post-test analysis, the prevalence of both relational and dysfunctional risks was expected to be higher in the control group than in the experimental group.

## 2. Materials and Methods

### 2.1. Design and Participants

To assess the effects of the Safety.net program, a repeated pre-posttest measure design was carried out with an intervention group (*n* = 120) and a control group (*n* = 45). The selection of the educational centers was carried out using nonprobability convenience sampling. The pre-test was carried out in December 2019 and the post-test in May–June 2020, during the COVID-19 lockdown. This study participants were 165 adolescents between 11 and 14 years old (M = 12.11, SD = 0.89) from 5 Spanish educational centers in three Spanish regions (Aragón, Asturias and Castilla-León). Table 1 shows the characteristics of the intervention and control groups by age, sex and academic year.

### 2.2. Multi-Risk Internet Prevention Program: Safety.Net

Safety.net is a prevention program aimed at jointly preventing various risks of the Internet, both relational (cyberbullying, sexting, online grooming, cyber dating abuse) and dysfunctional (problematic Internet use, nomophobia, Internet gaming disorder and online gambling disorder) in 11–14-year-old adolescents. Specifically, this program aims to primarily prevent the appearance and increase of these problems in adolescence [40].

Conceptually, this program is based on four theoretical frameworks: the theory of planned behavior [41], the social co-construction model [42], the cumulative risk model [43] and the empowerment theory [44]. Thus, first, from the theory of planned behavior, it is assumed that intentions influence the behavior, so if we change the motivational factors related to intentions, we can change the behavior. In this sense, throughout the Safety.net program, in addition to trying to change negative behaviors in the online context, it is also intended to change the motivations that influence the intentions to perform these inappropriate behaviors [41]. Second, based on the co-construction model, the Safety.net program also conceives that there is some parallelism in what happens in the online and offline contexts of adolescents’ lives [42]. Third, the program follows the cumulative risk model’s premises, considering that adolescents are often exposed to several risks at the same time and that such simultaneous exposure to various risks has worse consequences in their development than exposure to a single risk [43]. Finally, to provide adolescents with tools and resources they can use when they face the various risks of the Internet, this program is also based on the empowerment theory [44]. Concretely, the conceptual basis of the theory of planned behavior [41] and the social co-construction model [42] are taken into account in a transversal way in the explanation of concepts and activities proposed throughout the program Safety.net. The cumulative risk model [43] is taken into account above all in modules 2 and 3, which refer specifically to the Internet risks. Related to the empowerment theory [44], module 4 offers adolescents specific resources and skills to reinforce prevention and coping with risks on the Internet that the adolescents have seen previously in the program.

The program has followed the general lines of an instructional design; that is, it has sought to create some learning experiences that enable the acquisition of knowledge of the risks and the skills required to prevent them attractively [45]. In addition, the following principles have been followed for its construction: (1) to activate their prior knowledge in every session, creating a net (interconnected structure) by interweaving the risks; (2) to focus on the most important elements in each session; (3) to balance the cognitive content in and between sessions; (4) to use significant materials/activities [46].

Safety.net program is made up of 16 one-hour sessions (see Table 2), divided into 4 modules: (1) digital skills: the objective is to educate students about the characteristics of technologies that can carry risks and provide skills to prevent digital victimization and dysfunctional use of the Internet; (2) relational risks: this module aims to raise awareness about the seriousness of the risks arising from the relationships of students with other people through the Internet and to give them some advice on these problems; (3) dysfunctional risks: the purpose of this module lies on raising awareness about the seriousness of the risks derived from a dysfunctional use of the Internet and give them advice on how to safely use ICT’s; and (4) change of attitudes and cognitions: it aims to promote certain skills, competencies and abilities in adolescents so that they can better cope with the risks of the Internet. Each session’s specific structure is information about the risk or the construct, awareness, an individual or group activity that seeks to make the necessary changes in their attitude and behavior, offer recommendations labeled as “cyber tips” and group reflection on what they have learned. In addition to these sessions, the program includes two other specific sessions for the empirical assessment of the effectiveness of the program (pre-test and post-test). It is important to note that modules 2 and 3 and module 1 are the essential structure of the program. Module 4 complements them by reinforcing psychosocial skills that can help them prevent cyber risks that have been addressed throughout the program. Furthermore, each of the mentioned Internet risks is covered in a single session. However, the program tries to work on all the risks in an interconnected way, supporting some sessions on previous ones following an instructional design.

### 2.3. Instruments

To evaluate the effect of the Safety.net program, the Spanish version of the following questionnaires for relational and dysfunctional risks were administered.

Cyberbullying Triangulation Questionnaire (CTQ) [47]. For this study, the cybervictimization dimension of this questionnaire was used. This dimension has 9 items assessing the level of cybervictimization (e.g., “I have been sent some threatening or insulting messages.”). The answers of this Likert-type scale range from 0 (never) to 4 (almost every week). Cronbach’s alpha coefficient was 0.70 (pretest) and 0.71 (posttest).

Questionnaire for Sexual Requests and Interactions with Adults [48]. Through 10 items, it evaluates both sexual solicitation and interaction from an adult to a minor. Both are part of the online grooming process (e.g., “An adult has asked me to have cybersex—for example, through a webcam”). The answers to these items range from 0 (never) to 3 (6 or more times). Cronbach’s alpha coefficient was 0.80 (pretest) and 0.70 (posttest).

Victimization Scale of the Online Partner Abuse Questionnaire [7]. This scale consists of 11 items that assess control behaviors and direct aggression received by the partner through the Internet and smartphone (e.g., “He/she has called or messaged me too often to ask where I was and with whom”). The answers range from 0 (never) to 3 (almost always). Cronbach’s alpha coefficient was 0.74 (pretest) and 0.70 (posttest).

Additionally, three of the items of the Sexting Questionnaire created by [49] were used. These are focused on the sending of sexual or private information or messages to three potential recipients: (1) partner, (2) friend or acquaintance, and (3) someone they have met on the Internet, but not in person (e.g., “You have sent some information, photos or videos to your partner/friend/stranger with private or sexual contents about yourself”). The answers range from 0 (never) to 4 (7 or more times). The original scale has suitable reliability and validity indicators [49].

Generalized and Problematic Internet Use Scale (GPIUS2) [50]. It evaluates using 15 items the compulsive use, preference for online social interaction, cognitive preoccupation, mood regulation and negative consequences (e.g., “It is difficult for me to control the amount of time I spend using the Internet”). The answers of this scale range from 0 (totally disagree) to 5 (totally agree). Cronbach’s alpha coefficient was 0.88 (pretest) and 0.94 (posttest).

Nomophobia Questionnaire (NMP-Q) [14]. It has 20 items, which evaluate irrational, disproportionate fear of the possibility of not being able to use the mobile phone or running out of signal and/or battery (e.g., “I wouldn’t feel well if I weren’t able to access information through my smartphone at any time”). The answers of this scale range from 0 (totally disagree) to 6 (totally agree). Cronbach’s alpha coefficient was 0.95 (pretest) and 0.96 (posttest).

Internet Gaming Disorder Scale-Short Form (IGDS9-SF) [11]. This scale is composed of 9 items, and it evaluates Internet gaming disorder in adolescents following the DSM-5 criteria (e.g., “Do you systematically fail when you try to control or end your activities with video games?”). The answers range from 0 (never) to 4 (very often). Cronbach’s alpha coefficient was 0.83 (pretest) and 0.90 (posttest).

Online Gambling Disorder Questionnaire (OGD-Q) [13]. This scale has 11 items, which evaluate online gambling disorder in adolescence (e.g., “Do you feel nervous, irritated or angry when you try to reduce or quit online gambling?”). The answers of this scale range from 0 (never) to 4 (every day). Cronbach’s alpha coefficient was 0.95 (pretest) and 0.96 (posttest).

### 2.4. Procedure

First, contact was made with an educational institution that groups together numerous educational centers in Spain. Once a collaboration agreement had been signed, several schools were chosen on account of their convenience. The necessary institutional permits were obtained, and the parents or guardians of the children who took part in intervention groups had to sign an informed consent form. For the students in the control groups, passive consent was chosen (since this group only participated in administering the questionnaires and not the program). This passive consent was sent through the official way established by each school and, when the families refused to give their consent, it was to be sent back signed to the school so that the minor would not take part in the study (less than 1%).

The teachers in each school were responsible for the administration of the program. All the teachers who took part received specific training (30 h) to implement the program. Such training was given on an online platform containing all the training resources (materials, activities, videos, documentation, etc.). Everything was at their disposal on the website of the program www.programasafety.net (accessed on 1 March 2021). The program was implemented from December 2019 to March 2020 during tutorship hours. Due to the state of alert decreed by the Spanish government on March 11th on account of COVID-19, module 4 of the program could not be implemented. It is a supplementary module that reinforces the contents taught in the three previous ones.

The pre-test evaluation took place in an online format in the computer science classrooms of each school under the supervision of each course tutor. In the post-test phase, due to the lockdown situation, the questionnaires were filled out by the students from home during school hours under the teachers’ online supervision. The teachers reminded the students at all stages that there were no right or wrong questions and that they should answer honestly. The Survey Monkey© (SVMK Inc., San Mateo, CA, USA) survey platform was used.

This study was approved by the Ethics Committee of the International University of La Rioja PI: 004/2019). The study was brought to the attention of any juvenile prosecution services involved to follow the legal procedure for such cases.

### 2.5. Statistical Data

The statistical analyses were carried out using the Statistical Package for the Social Sciences (SPSS) 25 (IBM^®^, Armonk, NY, USA) program. First, Cronbach’s alpha was analyzed to check the internal consistency of each of the instruments used in this study. The total scores of each construct were obtained by adding the scores of all of its items.

To evaluate the effects of the Safety.net program on each variable of the study, the repeated measures analysis of variance (ANOVA 2×2) was used with an inter-group factor (intervention group and control group) and an intra-subject factor (before and after the program: pre-test and post-test). A value of *p* ≤ 0.05 was considered significant. The use of this statistical test is recommended when the selected groups are natural groups and not equal in the initial situation–pre-test– in at least one variable. An intervention group was used because the sample was not randomized [51]. The intervention’s effectiveness was tested with the interaction between the group (intervention or control) and the time (pre-test or post-test) in each repeated measure. In addition, the partial eta squared statistic has been used as an indicator of the magnitude of the effect of the contrasting performed. This effect was considered small when η^2^ < 0.06, medium when η^2^ ≥ 0.07 ≤ 0.14 and big when η^2^ > 0.14 [52].

## 3. Results

### 3.1. Sociodemographic Data of the Intervention and Control Group

To evaluate the effects of the Safety.net program, 120 adolescents were assigned to the intervention group, where the Safety.net program was implemented during school hours, and 45 to the control group, where the program was not implemented. The distribution of both groups by sex, age and academic year is shown in Table 1. No significant differences have been found between the intervention group and the control group, who are similar in terms of age (*t* = 1.60, *p* = 0.110) and sex (*χ^2^*(1, 165) = 1.03, *p* = 0.369). Regarding the intervention group, no significant differences have been found in any of the pre-test study variables, except online grooming, where the intervention group has slightly lower scores (*t* = 2.11, *p* = 0.037).

### 3.2. Effects of the Safety.Net Program on Relational Internet Risks

Table 3 shows how cybervictimization, although without a significant effect on the time x group interaction, does have significant main effects, both regarding time (*F*(1, 163) = 13.44, *p* < 0.01,) and the group (*F*(1, 163) = 5.64, *p* = 0.02), with medium (η^2^ = 0.076) and small (η^2^ = 0.012) effects, respectively. In the online grooming variable, a significant group x time interaction effect was found (*F*(1, 163) = 6.20, *p* = 0.01), with a small effect (η^2^ = 0.04).

As regards sexting and cyber dating abuse, no significant main or interaction effects were found, as shown in Table 3 and Figure 1. Concretely, sexting shows a tendency to increase in both the intervention group and the control group. The cyber dating abuse shows a decrease in both groups (intervention and control) between both measuring times even if there is no significant effect of time or group (nor of the interaction of both).

### 3.3. Effects of the Safety.Net Program on Dysfunctional Internet Risks

Regarding the dysfunctional Internet risks, the results show a significant effect of time x group interaction on problematic Internet use (*F*(1, 147) = 8.29, *p* = 0.005) and on Internet gaming disorder (*F*(1, 157) = 5.39, *p* = 0.02), with a small effect on both Internet risks (η^2^ = 0.05 and 0.03, respectively). As shown in Table 4, these problems have a higher increase in the control group than in the intervention group.

At the same time, a significant interaction effect was found on the nomophobia variable (*F*(1, 147) = 19.90, *p* = 0.01), with a medium effect (η^2^ = 0.12). In this case, the results evidenced that, whereas the intervention group’s scores are lower, the control group’s scores regarding this Internet risk have increased. No significant main or interaction effects have been found regarding online gambling disorder (Table 4 and Figure 2).

## 4. Discussion

This study is the first to present a multi-risk Internet prevention program to answer the pressing need for prevention given the polyhedral, ever-changing reality that affects adolescents. The program was designed to address four relational Internet risks (cyberbullying, sexting, online grooming, and cyber dating abuse) and four dysfunctional Internet risks (problematic Internet use, nomophobia, Internet gaming disorder, and online gambling disorder). Furthermore, the program is characterized by its minimal intervention (a one-hour session to address each risk) using a network instructional design that seeks to maximize the connections between risks and their prevention, generating significant changes between pre- and post-implementation. For these reasons, the program is a remarkable contribution to the field of prevention, especially considering the worrying growth and concurrence of many risks of the Internet [16,17,19].

Regarding relational Internet risks, the control group showed higher rates of cybervictimization compared to the intervention group. This was also true for previous programs, such as the cyber-friendly program, Prev@cib, and Cyberprogram 2.0 [27,31,32]. However, it is important to note that an increase in the cybervictimization rate was seen in both the control and intervention groups in the post-test analysis compared to the pre-test analysis. In this regard, the Safety.net program was able to buffer the natural increase in these intervention groups’ problems compared to the control group [21]. This may be even more relevant considering the COVID-19 pandemic and the resulting lockdown. In this context, adolescents may have been exposed to more online risks and content related to hatred or cyberbullying [53], sexting [54], or problematic use of the Internet or mobile phones [55,56]. There are several possible explanations for increased vulnerability to online risks during a lockdown. One such potential explanation lies in changes in adolescents’ lifestyles, characterized by decreased physical activity and increased levels of Internet and mobile phone use [57,58]. Additionally, this vulnerability could be higher due to associations found between problematic Internet-related behaviors and psychological distress during the COVID-19 pandemic [56,58]. Another potential explanation is the combination of decreased parental supervision of Internet use and interferences in parent–child relationships caused by stress and the changing work situations of families during confinement [59], which has been identified as an increasing factor of online risks [60,61,62].

This program also served as a buffer for online grooming; the adolescents in the control group demonstrated significantly increased post-test online grooming rates, while these rates decreased slightly in the intervention group. These results are similar to those of other programs that have shown a reduction in reciprocity between sexual solicitation and sexual interaction with adults in one session [29,33]. The purpose of the Safety.net program is to implement primary prevention strategies in early adolescence, which may explain why there were no significant results concerning cyber dating abuse. Pre-adolescents have fewer intimate relationships than adolescents and, when these relationships do exist, they are usually sporadic and short [63]. In addition, the prevalence of sexting increased overall; some studies have suggested that this practice was reinforced during this period because of the impossibility of physical, sexual contact [54,64].

The results regarding problematic Internet use and Internet gaming disorder showed a buffering effect in the intervention group compared to the control group. It should be noted that the evaluation of Internet gaming disorder is carried out from a clinical perspective [11]; therefore, the changes observed in this study are particularly interesting, as they are associated with clinical criteria. In this category, the most relevant changes were associated with the nomophobia construct; there was a significant score reduction between pre-test and post-test in the intervention group, as well as a notable difference from the control group in the post-test. Such reduction may be due to the fact that nomophobia is a problem that particularly affects 12- to 15-year-old adolescents [14], and the Safety.net program appropriately addresses the age group that will potentially benefit the most from it. No effects were found concerning online gambling disorder, but this may be due to the fact that the sample was primarily pre-adolescent (under 14 years of age); gambling prevalence at this age is minimal, as is the presence of diagnostic criteria for online gambling disorder [13]. A particularly promising option for future studies is to include “loot boxes” when addressing gambling disorder prevention [65].

It should be noted that this study was considered exploratory, particularly because of the impossibility of completing the fourth module (focused on cognitions and attitudes changes) and the lockdown caused by the COVID-19 pandemic. The first three modules are essential for the program, but the fourth one (while it is an important complement) is unnecessary. Nevertheless, the pandemic may have turned out to be a very pertinent unintentional context for analyzing the program’s effectiveness. Lockdowns and social distancing measures have increased people’s exposure to the Internet in general. Since the post-test was evaluated during the lockdown in Spain, such evaluation may have been influenced by a situation of overexposure to the Internet and its potential consequences [53,66]. The lockdown may have been a singular ecological context from which the participants of the intervention group may have benefited in contrast with the control group (most risks significantly increased during this period). It should also be noted that although the program acted as primary prevention (preventing the emergence of problems) for some adolescents, for others, it may have had a secondary prevention effect (preventing situations from becoming worse in cases of adolescents who were already at a certain level of risk). All of these factors underscore the need to carry out further studies on the effectiveness of the program.

This study is not without its limitations. First, the results were based on self-reports, which may have included a level of response bias. Second, the pandemic context may have added additional variables to the T2 sample collection process. Third, there was high experimental mortality due to the extraordinary circumstances and the fact that families and course tutors found it difficult to offer the required contact and attention during the lockdown. The study’s exploratory nature and the sample forces us to be cautious with the interpretation and generalization of the results. In addition, the impossibility of implementing the fourth module (which has a complementary nature) because of the closing of schools during the state of alert in Spain is a limitation that should be considered. To confirm the program’s effectiveness, future studies should increase the number of participants in both the intervention and control groups, implement the entire program, and include a random assignment and an additional measure after the post-test.

This study has important implications. It offers a preventive program that addresses many risks of the Internet in minimal sessions. The program is intended to be implemented within the framework of tutorship action. The materials found in the program would provide any guidance department, course tutor, or teacher with numerous resources with which to promote its application in class. In the coming years, at least until society recovers from the COVID-19 pandemic situation, the need to prevent problems of this nature is likely to increase. This is particularly true in the potential case of future lockdowns (especially if schools must remain closed and children need to learn at home).

## 5. Conclusions

The preliminary finds of this study suggest that Safety.net can be an effective program, especially to prevent online grooming, problematic Internet use, nomophobia and Internet gaming disorder in early adolescence.

## Figures and Tables

**Figure 1 ijerph-18-04249-f001:**
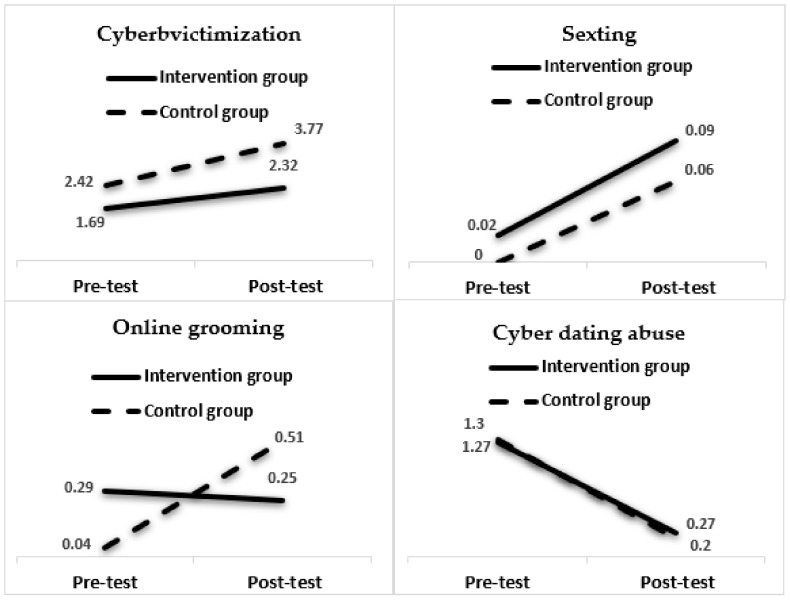
Means obtained by the groups (intervention and control) in online Internet risks.

**Figure 2 ijerph-18-04249-f002:**
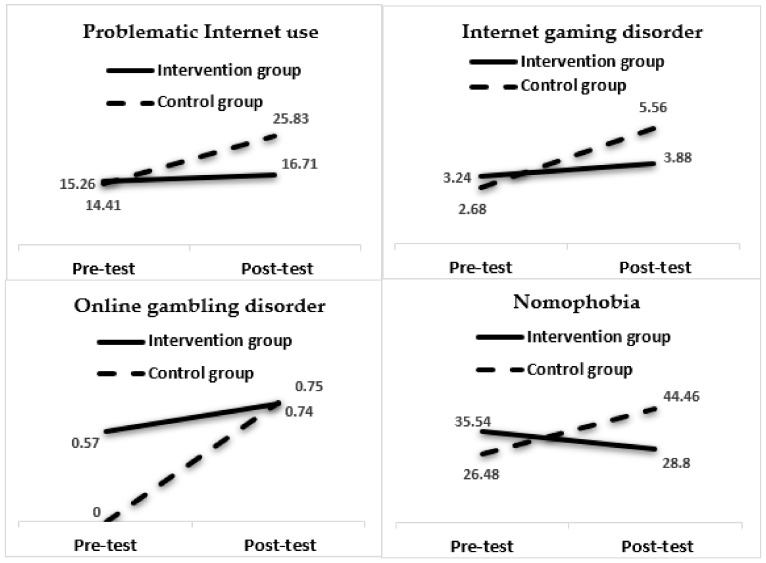
Means obtained by the groups (intervention and control) in dysfunctional Internet risks.

**Table 1 ijerph-18-04249-t001:** Characteristics of the intervention and control groups by age, sex and academic year: frequency and (percentage).

Variables	Intervention Group (*n* = 120)	Control Group (*n* = 45)
Age M (SD)	12.18 (0.81)	11.93 (1.07)
Sex		
Boys	43 (26.1%)	20 (12.1%)
Girls	77 (46.7%)	25 (15.2%)
Academic year		
Sixth grade of primary education	26 (15.8%)	20 (12.1%)
First grade of compulsory secondary education (CSE)	53 (32.1%)	12 (7.3%)
Second grade of compulsory secondary education (CSE)	41 (24.8%)	13 (7.9%)

Age (M = arithmetic mean; SD = standard deviation); age, sex and academic year.

**Table 2 ijerph-18-04249-t002:** Structure of sessions and modules of the Safety.net program.

	Modules	Sessions	Activities
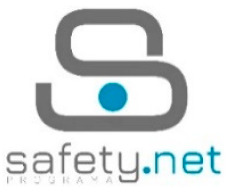	Module 1: Digital skills	Session 1: Netiquette	Activity 1: Netiquetting
		Session 2: Audience and digital footprint	Activity 2: Send to all
		Session 3: False profiles	Activity 3: False profiles
		Session 4: Privacy	Activity 4: What is known of me
	Module 2: Relational risks	Session 5: Cyberbullying Session 6: Sexting Session 7: Online grooming Session 8: Cyberdating abuse	Activity 5: Moni, the monkey Activity 6: True or false Activity 7: Find the 7 differences Activity 8: Reflecting on cyberdating abuse
	Module 3: Dysfunctional risks	Session 9: Problematic Internet use	Activity 9: Connected
		Session 10: Video games and online gambling	Activity 10: Battle of kings
		Session 11: Nomophobia and FoMO	Activity 11: Disconnected
	Module 4: Change of attitudes and cognitions	Session 12: Self Online	Activity 12: My Instaguay
		Session 13: Anonymous heroes	Activity 13: I will help you
		Session 15: Online emotional intelligence	Activity 15: Thought-action-emotion
		Session 16: Challenge	Activity 16: I think, so I challenge

**Table 3 ijerph-18-04249-t003:** Intergroup effects and repeated measures analysis of variance (ANOVA 2×2) in relational Internet risks.

Variables	M (SD)			*F* (η^2^)		
	Group	Pre-Test	Post-Test	Time Effect	Group Effect	Interaction Effect
Cyber victimization	Intervention	1.69 (2.35)	2.32 (2.89)			
	Control	2.42 (3.93)	3.77 (3.98)			
				13.44 *** (0.08)	5.64 * (0.01)	1.77 (0.02)
Sexting	Intervention	0.02 (0.15)	0.09 (0.45)			
	Control	0.00 (0.00)	0.06 (0.25)			
				3.13 (0.02)	0.47 (0.00)	0.00 (0.00)
Online grooming	Intervention	0.29 (1.33)	0.25 (1.11)			
	Control	0.04 (0.20)	0.51 (1.29)			
				4.65 * (0.03)	0.00 (0.00)	6.20 * (0.04)
Cyber dating abuse	Intervention	1.27 (1.95)	0.27 (0.64)			
	Control	1.30 (2.86)	0.20 (0.42)			
				4.03 (0.18)	0.002 (0.00)	0.009 (0.00)

η^2^ = partial eta squared * *p* < 0.05; *** *p* < 0.001.

**Table 4 ijerph-18-04249-t004:** Intergroup effects and repeated measures analysis of variance (ANOVA 2×2) in dysfunctional Internet risks.

Variables	M (SD)			*F* (η^2^)		
	Group	Pre-Test	Post-Test	Time Effect	Group Effect	Interaction Effect
Problematic Internet use	Intervention	15.26 (13.21)	16.71 (15.21)			
	Control	14.41 (11.85)	25.83 (19.01)			
				13.84 *** (0.09)	4.17 * (0.03)	8.29 ** (0.05)
Nomophobia	Intervention	35.54 (27.44)	28.80 (27.02)			
	Control	26.48 (25.12)	44.46 (32.20)			
				4.11 * (0.03)	0.59 (0.00)	19.90 *** (0.12)
Internet gaming disorder	Intervention	3.24 (4.78)	3.88 (4.84)			
	Control	2.68 (4.67)	5.56 (7.04)			
				1.36 (0.01)	5.23 * (0.03)	5.39 * (0.03)
Online gambling disorder	Intervention	0.57 (2.62)	0.74 (3.48)			
	Control	0.00 (0.00)	0.75 (3.49)			
				1.65 (0.01)	0.65 (0.01)	0.48 (0.00)

η^2^ = partial eta squared * *p* < 0.05; ** *p* < 0.01; *** *p* < 0.001.

## Data Availability

The data have not been incorporated into the study or evaluated by the reviewers. They will not be provided at this stage.

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
