# Peer review of "Safety.Net: A Pilot Study on a Multi-Risk Internet Prevention Program"

_ijerph, 2021, doi:10.3390/ijerph18084249_

Round 1
Reviewer 1 Report
Abstract: First two sentences should be written in a more professional format. I was a little unclear on the sentence “A one-hour session is dedicated to each risk” and needs more detail. Other than that, the abstract is clear and concise, just like a good abstract should be.
- Introduction: Remove all use of first-person.
- Material and Methods:
- 1: Remove extra period at the end of the sentence. Table 1 does not have the information that it says it should (looks like Table 2 does, however). Should be labeled correctly and should be moved up underneath section 2.1.
- 2 Multi-risk: The TPB needs more explanation and detail. It’s explained incorrectly in the text. Remove “and fourthly”. Overall, a better explanation is needed in how this program is ingrained and guided by these four theories (including their constructs). The description of “digital skills” sounds as though the program is providing skills to prevent digital victimization. With the description provided, the name should be changed to “digital risks.” Change sentence that says “It is important to note that modules 2 and 3, as well as module 1…” to “Modules 1-3 are the essential structure…”.
- Table 1: remove question mark from “true or false” on Activity 6 to keep consistency of not using punctuation marks. Isn’t Module 3 supposed to have “Dysfunctional Risks” underneath it instead of “Relational Risks”?
- 2 Instruments: 2.2 used twice; here and the section before. Omit first sentence as it is repetitive from the previous section and irrelevant to this section. More detail needed about which instruments were used for which module sessions.
- 4: Okay
- Results:
- 1: Okay
- Table 2: right justify the text instead of centering it.
- 2: Paragraphs should have a minimum of three sentences, not one. More detail needed to explain Table 3 and Figure 1.
- Table 3: Okay
- Figure 1: Sexting increased over time for both groups?
- 3: More detail needed in text to explain data in Tables 4 and Figure 2.
- Table 4: Okay
- Figure 2: Okay
- Discussion: Good explanation of why cybervictimization and sexting may have increased due to COVID lockdown. This section was the strongest, overall.
- Conclusions: Okay.
References: References appear to be consistent with the IOS 4 rules provided on the example on IJERPH website.
Comments: Overall, the text of the manuscript needs work and more attention to detail. The following changes are necessary:
- Remove and substitute for all forms of “it” which was found throughout the manuscript.”
- Omit all use of first-person in professional writing.
- Fix grammatical errors and typos noted throughout manuscript.
- Remove all unnecessary words.
Overall, the writing style and grammar needs work and attention, but the topic is very important and timely. Most importantly, the implications from this study are significant. I would love to see this study replicated, with all four modules, after the pandemic. However, given the increased use of the Internet and social media during the pandemic, it’s amazing to find an online program that can help decrease some of the risk factors of online victimization and problematic behaviors as well as increase protective factors. Every adolescent should have access to this program.
Author Response
Reviewer 1's comments
Abstract: First two sentences should be written in a more professional format. I was a little unclear on the sentence “A one-hour session is dedicated to each risk” and needs more detail. Other than that, the abstract is clear and concise, just like a good abstract should be.
Authors' responses -> Thank you for your overall assessment of the manuscript. We appreciate your helpful comments and have addressed most of them and the manuscript has improved. Your feedback on the importance of an Internet multi-risk prevention program is very important to us. At this moment, we are developing a second phase with 450 students in the intervention group and 400 in the control group which is developing normally. The research team is strongly committed to the project.
We welcome your comments on the abstract. We have incorporated the changes.
Abstract: Many programs exist to prevent bullying and cyberbullying. Nevertheless, despite evidence of the numerous overlapping risks of the Internet, programs that jointly and adequately address large sets of risks are not presently described in the scientific literature. The main objective of this study was to assess the effectiveness of the Safety.net program in a pilot sample. This program prevents eight Internet risks: cyberbullying, sexting, online grooming, cyber dating abuse, problematic Internet use, nomophobia, Internet gaming disorder, and online gambling disorder. The Safety.net program is made up of 16 sessions and 4 modules (digital skills, relational risks, dysfunctional risks, and change of attitudes and cognitions). Each session lasts one hour, but the program has a networked instructional design so that previous content can be recalled in later sessions.
Reviewer 1's comments:
Introduction: Remove all use of first-person.
Authors' responses -> The manuscript has been sent to an external translation company for English revision. We hope it has been significantly improved.
Occasionally, we use the first personal plural, but never the first personal singular. In any case, we have modified it as well.
Reviewer 1's comments
Material and Methods:
- 1: Remove extra period at the end of the sentence. Table 1 does not have the information that it says it should (looks like Table 2 does, however). Should be labeled correctly and should be moved up underneath section 2.1.
- Authors' responses -> Thank you for your comment. You are absolutely right. The tables were not well nominated. We have improved this point.
- 2 Multi-risk: The TPB needs more explanation and detail. It’s explained incorrectly in the text. Remove “and fourthly”. Overall, a better explanation is needed in how this program is ingrained and guided by these four theories (including their constructs). The description of “digital skills” sounds as though the program is providing skills to prevent digital victimization. With the description provided, the name should be changed to “digital risks.” Change sentence that says “It is important to note that modules 2 and 3, as well as module 1…” to “Modules 1-3 are the essential structure…”.
- Authors' responses -> We have explained better the theoretical frameworks on the Safety.net program, we have modified the explanation about TBP, and we have checked the typographic mistakes according to your suggestions.
Conceptually, this program is based on four theoretical frameworks: the theory of planned behavior [41], the social co-construction model [42], the cumulative risk model [43] and the empowerment theory [44]. Thus, firstly, from the theory of planned behavior, it is assumed that intentions influence on the behavior, so if we change the motivational factors related to intentions, we can change the behavior. In this sense, throughout the Safety.net program, in addition to trying to change negative behaviors in the online context, it is also intended to change the motivations that influence the intentions to perform these inappropriate behaviors. [41]. Secondly, on the basis of the co-construction model, Safety.net program also conceives that there is some parallelism in what happens in the online and offline contexts of adolescents' lives [42]. Thirdly, the program follows the premises of the cumulative risk model considering that adolescents are often exposed to several risks at the same time and that such simultaneous exposure to various risks has worse consequences in their development than exposure to a single risk [43]. Finally, in order to provide to adolescents with tools and resources they can use when they face the various risks of the Internet, this program is also based on the empowerment theory [44]. Concretely, the conceptual basis of the theory of planned behavior [41], and the social co-construction model [42] are taken into account in a transversal way in the explanation of concepts and activities proposed throughout the program Safety.net. The cumulative risk model [43] is taken into account above all in the module 2 and 3, which refer specifically to the Internet risks. Related to the empowerment theory [44], module 4 offers adolescents specific resources and skills to reinforce prevention and coping with risks on the Internet that the adolescents have seen previously in the program.
- Table 1: remove question mark from “true or false” on Activity 6 to keep consistency of not using punctuation marks. Isn’t Module 3 supposed to have “Dysfunctional Risks” underneath it instead of “Relational Risks”?
- Authors' responses -> You are absolutely right. It was a layout problem when using the IJERPH model. Thank you for your comment.
- 2 Instruments: 2.2 used twice; here and the section before. Omit first sentence as it is repetitive from the previous section and irrelevant to this section. More detail needed about which instruments were used for which module sessions.
- Authors' responses -> You are absolutely right. It was a layout problem when using the IJERPH model. We have eliminated the first two sentences. Thank you for your comment.
Reviewer 1's comments
Results:
- 1: Okay
- Table 2: right justify the text instead of centering it.
- 2: Paragraphs should have a minimum of three sentences, not one. More detail needed to explain Table 3 and Figure 1.
- Table 3: Okay
- Figure 1: Sexting increased over time for both groups?
- 3: More detail needed in text to explain data in Tables 4 and Figure 2.
- Table 4: Okay
- Figure 2: Okay
Authors' responses -> Thank you for your comments. We have modified the justification of table 2 and increased the text of the results for relational risks, but we did not want to be redundant with the information provided in the tables/figures.
Reviewer 1's comments
Discussion: Good explanation of why cybervictimization and sexting may have increased due to COVID lockdown. This section was the strongest, overall.
Authors' responses ->Thank you for your kind comment.
Reviewer 1's comments
Conclusions: Okay.
Authors' responses -> Thank you for your kind comment.
Reviewer 1's comments
References: References appear to be consistent with the IOS 4 rules provided on the example on IJERPH website.
Authors' responses -> The authors use the ZOTERO bibliography manager. For this we have downloaded the rules of the journal IJERPH and we have used them at all times.
We believe that this point is adequate as it is.
Reviewer 1's comments
Comments: Overall, the text of the manuscript needs work and more attention to detail. The following changes are necessary:
- Remove and substitute for all forms of “it” which was found throughout the manuscript.”
- Omit all use of first-person in professional writing.
- Fix grammatical errors and typos noted throughout manuscript.
- Remove all unnecessary words.
Overall, the writing style and grammar needs work and attention, but the topic is very important and timely. Most importantly, the implications from this study are significant. I would love to see this study replicated, with all four modules, after the pandemic. However, given the increased use of the Internet and social media during the pandemic, it’s amazing to find an online program that can help decrease some of the risk factors of online victimization and problematic behaviors as well as increase protective factors. Every adolescent should have access to this program.
Authors' responses -> As we have indicated in a previous comment, the authors have had the text professionally edited by a native speaker to improve the final text.
Reviewer 2 Report
The monuscript is very interesting and addresses topical issues, also in response to the spread of Covid-19 and the related hyper-connection problems. Some clarifications and requests for further information are proposed below.
Introduction. It is advisable to add hypotheses, theories and studies relating to the increase in the phenomena analyzed as a consequence of the spread of Covid-19 in the introductory paragraph and in the bibliography (the studies cited refer to the period prior to Covid-19, and it is assumed that things have gotten much worse over the past few months). Furthermore, greater clarity is required on the hypothesis and not only on the objectives of the manuscript. In particular, it is necessary to clarify the cause and effect relationships that could arise on the experimental group, in relation to the choice of methodology and tools adopted.
Methodology. It is suggested to:
- argue why a random sample was not created; explain why socio-demographic information was not collected also relating to the family context, such as the educational qualification and the profession of the parents, variables potentially influencing attitudes and behaviors;
- specify the reasons for choosing the specific (and not other) scales used to measure online grooming, problem internet use, nomophobia and internet gambling disorder in early adolescence;
- indicate all the procedures and measures adopted to minimize the risk of social desirability, especially in consideration of the questionnaires that students fill out independently at home, even if under the supervision of teachers; clarify the timing (pre / post) of the Safety.net program;
- deepen the analysis of the results in relation to the personal variables of the interviewees.Discussione
Discussion. It should be made clear that, before it can be stated that Safety.net can be a particularly effective program for preventing online grooming, problematic Internet use, nomophobia and Internet gaming disorder in early adolescence, It’s necessary:
- repeat the tests on the experimental group after a longer time;
- increase the number of participants and build a random sample of schools;
- discuss the risks related to the fact that the group may have offered answers related to the context situation, the presence of teachers and the social desirability, already strongly perceived in pre-adolescence.
Author Response
Reviewer 2's comments
The monuscript is very interesting and addresses topical issues, also in response to the spread of Covid-19 and the related hyper-connection problems. Some clarifications and requests for further information are proposed below.
Authors' responses -> Thank you for your overall assessment and interest in the topic of study. We believe that prevention programs that jointly address numerous risks better respond to the reality to which adolescents are exposed.
Reviewer 2's comments
Introduction. It is advisable to add hypotheses, theories and studies relating to the increase in the phenomena analyzed as a consequence of the spread of Covid-19 in the introductory paragraph and in the bibliography (the studies cited refer to the period prior to Covid-19, and it is assumed that things have gotten much worse over the past few months). Furthermore, greater clarity is required on the hypothesis and not only on the objectives of the manuscript. In particular, it is necessary to clarify the cause and effect relationships that could arise on the experimental group, in relation to the choice of methodology and tools adopted.
Authors' responses -> The authors have considered that the introduction should focus on analyzing individually and jointly the risks included in Safety.net, in addition to the existing prevention programs. We believe that the COVID-19 situation is a point that needs to be discussed in depth, as it has particularly affected us in the piloting of this study. Thus, we have included some recent studies in the discussion on this point.
"Regarding relational Internet risks, the control group showed higher rates of cybervictimization compared to the intervention group. This was also true for previous programs, such as the Cyber Friendly program, Prev@cib, and Cyberprogram 2.0 [27,31,32]. However, it is important to note that an increase in cybervictimization rate was seen in both the control and intervention groups in the post-test analysis compared to the pre-test analysis. In this regard, the Safety.net program was able to buffer the natural in-crease in these problems in the intervention group compared to the control group [21]. This may be even more relevant considering the COVID-19 pandemic and the resulting lockdown. In this context, adolescents may have been exposed to more online risks and content related to hatred or cyberbullying [53], sexting [54], or problematic use of internet or mobile phones [55,56]. There are several possible explanations for increased vulnerability to online risks during a lockdown. One such potential explanation is changes in adolescents’ lifestyles, characterized by a decrease in their physical activity and increased levels of Internet and mobile phone use [57,58]. Additionally, this vulnerability could be higher due to associations found between problematic Internet-related behaviors and psychological distress during the COVID-19 pandemic [56,58]. Another potential explanation is the combination of decreased parental supervision of Internet use and interferences in parent-child relationships caused by stress and the changing work situations of families during confinement [59], which has been identified as an increasing factor of online risks [60–62]..."
We authors have discussed among ourselves the point of the hypotheses on several occasions. We understand your comment, but we finally considered writing it in its current format because we do not have other similar experiences in which in such a small number of sessions so many risks have been incorporated. Each risk is worked on in one session, even if the instructional design is done as a network. This is why we believe that our hypotheses are really in line with what is to be expected: that the intervention group does not increase in the post, while the control group does.
Reviewer 2's comments
Methodology. It is suggested to:
- argue why a random sample was not created; explain why socio-demographic information was not collected also relating to the family context, such as the educational qualification and the profession of the parents, variables potentially influencing attitudes and behaviors;
- Authors' responses -> The problem of not having a random assignment has been incorporated into the limitations and future lines. The main reason was that the study was carried out in a specific educational institution and in the pilot we chose the centers for reasons of opportunity (proximity to the research team and interest in participating in the research initiative). In future studies, a larger sample and random assignment will be included.
- specify the reasons for choosing the specific (and not other) scales used to measure online grooming, problem internet use, nomophobia and internet gambling disorder in early adolescence;
- Authors' responses -> The research group uses in its studies all the questionnaires used, some of them created by the team since there were no questionnaires validated for a Spanish sample (such as the nomophobia questionnaire or the IGDS9-SF). In general, most of the questionnaires are used by several research teams, which allows for greater comparability of results (as in the case of the questionnaires on cyberbullying, online grooming, cyber dating abuse, problematic Internet use, etc.). In the case of Online Gambling Disorder, the construct is very new and a questionnaire was recently created and validated by the research team.
- indicate all the procedures and measures adopted to minimize the risk of social desirability, especially in consideration of the questionnaires that students fill out independently at home, even if under the supervision of teachers; clarify the timing (pre / post) of the Safety.net program;
- Authors' responses -> This is an interesting comment. Certainly, as in most studies with self-administered instruments, it is very difficult to reduce social desirability. These studies are conducted in a controllable, anonymous and supervised environment. We have added an instruction that all teachers carried out. In the post-test we were unable to implement any additional measures. This point is discussed in the limitations section.
- deepen the analysis of the results in relation to the personal variables of the interviewees.Discussione
- Authors' responses -> Information was collected on the studies and professional activity of the students' families. We agree with you that these are measures that should be analyzed in subsequent studies, but in this pilot we did not consider including them and only focusing on the first results of the Program.
Reviewer 2's comments
Discussion. It should be made clear that, before it can be stated that Safety.net can be a particularly effective program for preventing online grooming, problematic Internet use, nomophobia and Internet gaming disorder in early adolescence, It’s necessary:
- repeat the tests on the experimental group after a longer time;
- increase the number of participants and build a random sample of schools;
- discuss the risks related to the fact that the group may have offered answers related to the context situation, the presence of teachers and the social desirability, already strongly perceived in pre-adolescence.
Authors' responses -> Currently, the conclusion of the study is written emphasizing the exploratory nature of the study and indicating that "Safety.net can be an effective...".
We are aware of the scope of the study and the need for further work along these lines. As we have indicated to reviewer 1, we are now conducting a new phase where we have randomized and have 450 participants in the experimental group and another 450 in the control group.
We have included part of his comment in the future lines section (to make it more explicit).
Reviewer 3 Report
The submitted manuscript meets all the requirements of research quality in terms of form and substance. It presents a multi-risk Internet prevention programme to address the needs of adolescents in the face of Internet-related risks. I believe that the contribution of this work is of high quality. My decision is therefore to accept the work.
Author Response
Authors' responses -> We are very grateful for your overall assessment of the work. Thank you for your time in reading and evaluating it.
Reviewer 4 Report
The objective of the article is to assess the effectiveness of the Safety.net program in a pilot sample (165 adolescents between 11 -14 years-old). Its main results show improvements in online grooming, problematic Internet use, Internet gaming disorder and nomophobia. Therefore, suggest that the Safety.net program is an effective program that prevents the increase of most of the risks and reduces some of them.
The paper is well written and is composed and structured by all the parts of a scientific article needs.
An experiment can include several experimental groups at the same time. Clarify if is not the case?
Page1/Line 1: Where is “The massive, daily use of information…” I suggest “The massive and daily use of information and communication technologies (ICTs)…”
Introduction:
If possible, clarify better the distinguish between control group and experimental group to non-specialists (obvious this is not mandatory).
Page 2: Use quotation marks in “Convivir en un mundo real y digital” [25] and “Cibermentores” [26]. The same to all similar expressions…
I recommend that part of section 4. Discussion moves on to section 5. Conclusions, also highlighting the expected future work.
Author Response
Reviewer 4's comments
The objective of the article is to assess the effectiveness of the Safety.net program in a pilot sample (165 adolescents between 11 -14 years-old). Its main results show improvements in online grooming, problematic Internet use, Internet gaming disorder and nomophobia. Therefore, suggest that the Safety.net program is an effective program that prevents the increase of most of the risks and reduces some of them.
Authors' responses -> We are very grateful for your overall assessment of the work. Thank you for your time.
Reviewer 4's comments
The paper is well written and is composed and structured by all the parts of a scientific article needs.
Authors' responses -> Thank you for your comment.
Reviewer 4's comments
An experiment can include several experimental groups at the same time. Clarify if is not the case?
Authors' responses -> We do not know if we have understood the comment correctly. There is only one experimental condition in the safety.net program.
Page1/Line 1: Where is “The massive, daily use of information…” I suggest “The massive and daily use of information and communication technologies (ICTs)…”
Authors' responses -> Thank you for your suggestion. We have incorporated it.
Introduction:
If possible, clarify better the distinguish between control group and experimental group to non-specialists (obvious this is not mandatory).
Authors' responses -> Technically, the study refers to an intervention group (it differs from the experimental group because we do not have a random assignment). In order to be more comprehensive, we have added this indication in section 2.5.
Page 2: Use quotation marks in “Convivir en un mundo real y digital” [25] and “Cibermentores” [26]. The same to all similar expressions…
I recommend that part of section 4. Discussion moves on to section 5. Conclusions, also highlighting the expected future work.
Authors' responses -> Thank you for your suggestion on the use of quotation marks. We have incorporated it. The structure of the discussion and conclusions follows the IJERPH format.